# Deformation-Thermal Co-Induced Ferromagnetism of Austenite Nanocrystalline FeCoCr Powders for Strong Microwave Absorption

**DOI:** 10.3390/nano12132263

**Published:** 2022-06-30

**Authors:** Ziwen Fu, Zhihong Chen, Rui Wang, Hanyan Xiao, Jun Wang, Hao Yang, Yueting Shi, Wei Li, Jianguo Guan

**Affiliations:** 1School of Science, Wuhan University of Technology, Wuhan 430070, China; ziwenfu@whut.edu.cn (Z.F.); 261222@whut.edu.cn (H.Y.); 261179@whut.edu.cn (Y.S.); 2State Key Laboratory of Advanced Technology for Materials Synthesis and Processing, Wuhan University of Technology, Wuhan 430070, China; 303565@whut.edu.cn (R.W.); hanyan_xiao@whut.edu.cn (H.X.); 303841@whut.edu.cn (J.W.); wellee@whut.edu.cn (W.L.)

**Keywords:** nanocrystalline alloy absorbents, induced ferromagnetism, ball milling, magnetic properties, microwave absorption

## Abstract

Nanocrystalline soft magnetic alloy powders are promising microwave absorbents since they can work at diverse frequencies and are stable in harsh environments. However, when the alloy powders are in austenite phase, they are out of the screen for microwave absorbents due to their paramagnetic nature. In this work, we reported a strategy to enable strong microwave absorption in nanocrystalline austenite FeCoCr powders by deformation-thermal co-induced ferromagnetism via attritor ball milling and subsequent heat treatment. Results showed that significant austenite-to-martensite transformation in the FeCoCr powders was achieved during ball milling, along with the increase in shape anisotropy from spherical to flaky. The saturation magnetization followed parabolic kinetics during ball milling and rose from 1.43 to 109.92 emu/g after milling for 4 h, while it exhibited a rapid increase to 181.58 emu/g after subsequent heat treatment at 500 °C. A considerable increase in complex permeability and hence magnetic loss capability was obtained. With appropriate modulation of complex permittivity, the resultant absorbents showed a reflection loss of below −6 dB over 8~18 GHz at thickness of 1 mm and superior stability at 300 °C. Our strategy can broaden the material selection for microwave absorbents by involving Fe-based austenite alloys and simply recover the ferromagnetism of industrial products made without proper control of the crystalline phase.

## 1. Introduction

With the development of wireless communication and radar technologies, microwave absorbing materials (MAMs) with strong absorption over different microwave bands and superior stability in harsh environments are more and more important to suppress microwave interference or to reduce the radar cross-section of objects [1]. Emerging MAMs are expected to be flexible in designing working bands [2,3] and have stable performance at elevated temperatures [4,5] or in corrosive environments [6,7]. These requirements are beyond the reach of traditional microwave absorbents with single composition, e.g., carbonyl iron [8,9]. Soft magnetic alloys are promising to meet these new requirements [10,11,12,13] since they can benefit from composition design. Many soft magnetic alloy absorbents were reported in the literature, ranging from binary alloys such as FeNi [14], FeCr [15], FeCo [16] and FeSi [17], to ternary/quaternary/pentanary/hexanary alloys such as FeSiAl [18], FeSiAlCr [19], NiCrAlY [20] and FeSiBPCuY [21]. By selecting different compositions, environmental stability, magnetic resonant frequencies, the working bands of alloy absorbents can be designed flexibly [22]. For example, varying the contents of Si and Al can change the magnetic resonant frequency of FeSiAl powders [23]. Adding Si, Al or Cr elements can significantly improve the anti-oxidation and anti-corrosion properties of alloy absorbents [24,25]. Adding Co element can obviously raise the Curie temperature of alloy absorbents [26].

Besides composition, the crystallographic structure of alloy absorbents is also vitally important for microwave absorption. Generally, Fe-based alloys show ferromagnetic body-centered cubic (BCC) structures at low temperature, whereas they exhibit paramagnetic face-centered cubic (FCC) structures at high temperature [27]. For example, FeSiAl alloy powders show ferromagnetic BCC and DO3 superlattice structures [6] at low and medium temperature, respectively, while they show paramagnetic FCC structures at high temperature. It is well known that the ferromagnetic phase could generate magnetic loss to the microwave, whereas the austenite phase shows negligible interaction with microwave. When austenite Fe-based alloys are produced purposely or accidentally without proper control of the technique parameters, e.g., 304 or 316L stainless steel, they are generally beyond the selection as microwave absorbents due to their paramagnetism nature.

Utilizing austenite Fe-based alloys as microwave absorbents is attractive since they can largely broaden the family of absorbents and endow new properties for microwave absorption. Interestingly, austenite structure in some Fe-based alloys was found to be sensitive to strain, especially in 304 and 316L stainless steel with a main composition of Fe, Cr and Ni [28,29]. By mechanical milling, powders of 304 stainless steel exhibited obvious transformation of the austenite phase to the martensite phase, leading to a significant increase in saturation magnetization (*M*_s_) from a negligible value to 130~140 emu/g [30]. The strain-induced martensite transformation also showed controllable kinetics with about 90% of phase transformation [31,32] and the formation of a nanocrystalline martensite phase with a dimension of 10~20 nm [33,34]. This phenomenon offers an opportunity to utilize austenite Fe-based alloy powders as microwave absorbents. Nevertheless, studies for strain-induced martensite transformation were focused on static magnetic properties, without considering their shape anisotropy, permeability dispersion and microwave absorption performances. Moreover, the *M*_s_ after phase transformation was not high enough, which limits the permeability and hence magnetic loss for microwave.

In this work, nanocrystalline FeCoCr powders of austenite structure are investigated as example materials since they are promising for use at high temperature and in corrosive environment. Deformation of the FeCoCr powders associated with their shape anisotropy was performed with attritor ball milling, followed by subsequent heat treatment. Paramagnetism-to-ferromagnetism transformation and its kinetics were investigated during ball milling and heat treatment. We obtained a significant increase in *M*_s_ from 1.43 to 181.58 emu/g by deformation-thermal co-induced ferromagnetism. A considerable increase in complex permeability, magnetic loss capability and hence microwave absorption of FeCoCr powders was achieved.

## 2. Materials and Methods

FeCoCr powder with 42.5 wt.% of Co, 15 wt.% Cr and residual Fe was prepared by gas atomization from Hunan Metallurgical Institute, Changsha, China. Ethanol of analytical purity was purchased from Sinopharm Chemical Reagent Co., Ltd., Shanghai, China, without further purification.

Attritor ball milling was employed to generate plastic deformation and shape anisotropy in FeCoCr powders. Raw FeCoCr powder of 10 g was added to the attritor ball milling jar with 100 mL of ethanol as the process control agent. The rotation speed was set as 10 Hz, with ball-to-powder weight ratio of 80:1. After being milled for different time, the resultant powders are separated from milling balls by a sieve and collected by a filter, followed by drying at 60 °C for 3 h. After ball milling, the powder was then thermally treated at different temperature for 0.5 h in vacuum. To reduce the permittivity of milled FeCoCr powders, part of the samples were then thermally treated in air.

To measure the electromagnetic parameters, FeCoCr powders were added to melted paraffin and mixed uniformly with mass ratio of 3.265:1. The mixture of FeCoCr powder and paraffin was then pressed into a coaxial ring sample with outer diameter of 7 mm, inner diameter of 3 mm and thickness of 2~3 mm. Vector network analyzer (Keysight N5224B, Santa Rosa, CA, USA) was used to measure the complex permittivity and permeability. The morphology of FeCoCr powders was investigated by scanning electron microscope (SEM, Hitachi S-4800, Tokyo, Japan). High-resolution microstructure of FeCoCr powder was studied by transmission electron microscopy (TEM, F200s, Hillsboro, OR, USA). For TEM observation, FeCoCr powders were cut into thin slices of nanoscale thickness by focused ion beam (FIB, FEI Strata 400S, Hillsboro, OR, USA). The compositions of FeCoCr were confirmed by energy dispersive spectrum (EDS) in both SEM and TEM. The crystalline structures of FeCoCr powders were investigated by X-ray diffraction (XRD, Bruker D8 Advance, Karlsruhe, Germany) and selected area electron diffraction (SAED) in TEM. The static magnetic properties of particles were measured using a vibrating sample magnetometer (VSM, Lakeshore 7404, Westerville, OH, USA). All the tests with VSM were carried out at least 3 times.

## 3. Results and Discussion

The deformation-thermal co-induced ferromagnetism associated with shape anisotropy and phase transformation was verified. The variation of shape anisotropy in the FeCoCr powders during ball milling can be seen in Figure 1a–c. In Figure 1a, the raw FeCoCr powders are mostly in spherical shape, with an average diameter of about 8 μm. Some non-spherical particles are also observed, which were produced due to collision and adhesion of droplets during the spraying process of gas atomization [35]. The inset in Figure 1a indicates the element composition of the FeCoCr particles, with a uniform distribution of each element and composition of Fe_42.5_Co_42.5_Cr_15_ (See Appendix A). With the increase in ball milling time, the shape of the FeCoCr powders gradually changed to flaky. After milling for 4 h, most of the powders were in flaky shape, see Figure 1b. The heat treatment did not cause changes in the shape of the FeCoCr powders, as shown in Figure 1c. The inset in Figure 1c indicates that the thickness of the FeCoCr powders was 2~5 μm after milling for 4 h. More details on the morphological change of the FeCoCr powders during milling can be seen in Appendix A. The phase transformation in the FeCoCr powders after ball milling and subsequent heat treatment were studied by XRD (Figure 1d–f). It can be seen from Figure 1d that the raw FeCoCr powders were composed of γ-austenite and ε-martensite phases, with γ-austenite as the dominating phase. Calculated according to Scherrer equation [36], the average grain size of γ-austenite in the raw FeCoCr powders was roughly 40 nm. After ball milling for 4 h (Figure 1e), obvious diffraction peaks of the α′-martensite phase are observed, with a minority of the ε-martensite phase. The average grain size of the generated α′-martensite phase was about 10 nm. After further heat treatment at 500 °C for 30 min in vacuum, only the α′-martensite phase can be observed, with an average grain size of about 30 nm. Therefore, deformation-thermal co-induced austenite-to-martensite transformation on nanoscale was achieved in the FeCoCr powders. The phase transformation of the FeCoCr powders after ball milling was further confirmed by TEM and SAED, see Appendix A. For the raw FeCoCr powders, periodical stripes with a spacing of 0.179 nm can be seen in high-resolution TEM image, corresponding to the ($\bar{2}$00) crystal plane of γ-austenite. The SAED pattern confirms a typical FCC structure with a crystal band axis of [11]. The presence of ε-martensite was not identified in the raw FeCoCr powders due to its low content and sampling of the TEM image. For the FeCoCr powders milled for 4 h, periodic streaks with intervals of 0.202 nm and 0.195 nm can be observed, corresponding to the (110) crystal plane of α′-martensite and the (002) crystal plane of ε-martensite, respectively. α′- and ε-martensite can be identified by the calibration of the diffraction rings in SAED, which is consistent with the XRD results. A typical nanograin of α′-martensite with a width of about 13 nm is shown in Appendix A. The deformation-thermal co-induced ferromagnetism in the FeCoCr powders was further verified by hysteresis loops, as is shown in Figure 1g–i. In Figure 1g, it can be seen that the *M*_s_ of the raw FeCoCr powders is negligibly small, with a value of only 1.43 ± 0.03 emu/g, showing paramagnetic status due to the paramagnetism of the austenite phase [37]. After ball milling for 4 h, partial ferromagnetism was achieved, with an *M*_s_ of 109.92 ± 3.93 emu/g, which was due to the formation of the ferromagnetic α′-martensite phase [37]. Full ferromagnetism was obtained after heat treatment of the milled FeCoCr powders, giving an *M*_s_ of 181.58 ± 5.74 emu/g. Hence, by the deformation-thermal co-induced ferromagnetism strategy, we managed to raise the *M*_s_ of the FeCoCr powders from 1.43 to 181.58 emu/g.

The evolution kinetics of *M*_s_ in the FeCoCr powders during deformation-thermal co-induced ferromagnetism was further investigated. Two-stage kinetics for *M*_s_ can be seen, i.e., (i) initial parabolic-like kinetics during ball milling associated with shape anisotropy and phase transformation (Figure 2a) and (ii) a subsequent increase in *M*_s_ associated with the heat treatment (Figure 2b). In the first stage, as shown in Figure 2a, *M*_s_ increased rapidly upon ball milling due to the sensitivity of the austenite structure to strain [32]; it then gradually increased with the extension of milling time until reaching a plateau at the milling time of 4 h. With the further extension of milling time, the *M*_s_ kept stable even at a milling time of 12 h. The parabolic evolution of *M*_s_ in the FeCoCr powder during deformation was similar to that in 304 and 316L stainless steel powders. The *M*_s_ in the FeCoCr powders increased from 1.43 ± 0.03 to 109.92 ± 3.93 emu/g after ball milling, a value slightly lower than that of 304 and 316L, which gave a value of 130~140 emu/g [30,32]. Since the 304 and 316L stainless steels mainly consist of Fe, Cr and Ni, the difference in *M*_s_ of the FeCoCr and stainless steel may be due to the existence of the Co element. The evolution kinetics of *M*_s_ for the FeCoCr powders over milling time is in accordance with XRD spectra, as shown in Appendix A showing the gradual transformation of the austenite phase to the martensite phase. In the second stage, as shown in Figure 2b, when heat treatment was introduced after ball milling, a significant increase in *M*_s_ in the FeCoCr powders was achieved at temperature up to 500 °C, whereas it decreased at higher temperature. From Figure 2b, it can be seen that the raw FeCoCr powders maintained their paramagnetism over heat treatment at temperature up to 900 °C, showing the highest *M*_s_ of only 6.78 ± 0.07 emu/g. The change of *M*_s_ for the raw FeCoCr powders after heat treatment at different temperature is in accordance with XRD spectra, see Appendix A showing a stable austenite phase. When the FeCoCr powders were milled for 0.5 h and heat-treated, the *M*_s_ gradually increased from 50.12 ± 2.23 emu/g at room temperature to 83.50 ± 1.29 emu/g at 500 °C; it decreased at higher temperature and reduced to 1.50 ± 0.03 emu/g after heat treatment at 900 °C. When the FeCoCr powders were milled for 4 h and heat-treated, the *M*_s_ increased from 109.92 ± 3.93 emu/g at room temperature to the maximum value of 181.58 ± 5.74 emu/g at 500 °C. Heat treatment at higher temperatures led to the descent of *M*_s_. The synergistic effect between deformation and heat treatment can be seen when comparing the maximum *M*_s_ with that at room temperature. Comparing the change of *M*_s_ in the raw FeCoCr powders and those after milling for 0.5 h and 4 h, it can be seen that heat treatment caused an increase in *M*_s_ by 5.35, 33.38 and 68.43 emu/g, respectively. Therefore, the introduction of strain by deformation is the prerequisite for further phase transformation; high-level deformation, i.e., longer ball milling time, assists martensite transformation during heat treatment.

Different from *M*_s_, the coercivity (*H*_c_) showed the opposite behavior (see Appendix A), i.e., it gradually decreased with the extension of milling time and showed a minimum value during subsequent thermal treatment. The decrease in *H*_c_ during deformation-thermal co-induced ferromagnetism may be attributed to the mutual exchange coupling of the martensite phase formed during milling [31]. At short milling time, the content of the martensite phase was low, and the magnetic domains were well separated from each other, leading to weak coupling interaction and high *H*_c_. With the increase in the content of the martensite phase, the ferromagnetic coupling was enhanced, which resulted in the reduction of *H*_c_. Further heat treatment below 500 °C caused an extra reduction of *H*_c_ due to the reduction of defects. At temperature higher than 500 °C, *H*_c_ gradually increased due to the inverse transformation of the martensite phase to the austenite phase.

The kinetic behavior of the milled FeCoCr powders during subsequent heat treatment was investigated by XRD. In Figure 2c, XRD spectra for the FeCoCr powders milled for 0.5 h and heat-treated at different temperature are shown. They indicate that after milling for 0.5 h, the α′-martensite phase, γ-austenite and ε-martensite phases existed simultaneously, giving an *M*_s_ of 50.12 ± 2.23 emu/g. When the temperature increased to 300 °C, mainly α′-martensite and ε-martensite phases were observed, resulting in an increase in *M*_s_ to 60.31 ± 3.57 emu/g. At 500 °C, obvious diffraction peaks of the α′-martensite phase were observed, indicating the massive transformation of the α′-martensite phase, which gave a maximum *M*_s_ of 83.50 ± 1.29 emu/g. At higher temperature, diffraction peaks of the γ-austenite phase become more and more obvious, indicating the inverse transformation of γ-austenite from α′-martensite. XRD spectra of the FeCoCr powders after milling for 4 h and subsequent heat treatment at different temperature are shown in Figure 2d. Compared to the XRD spectrum of the FeCoCr powders milled for 0.5 h, obvious diffraction peaks of the α′-martensite phase can be seen for those milled for 4 h at room temperature, with weak diffraction peaks of the γ-austenite phase; this is corresponding to the tendency of *M*_s_ over milling time. With the increase in heat treatment temperature, diffraction peaks of the α′-martensite phase were getting narrower and sharper, while diffraction peaks of the γ-austenite and ε-martensite disappeared gradually. After heat treatment at 500 °C, only the α′-martensite phase can be observed; this leads to an *M*_s_ of as high as 181.58 ± 5.74 emu/g. Heat treatment at higher temperature caused the appearance of γ-austenite and hence the reduction of *M*_s_. At 700 °C, γ-austenite dominated the main phase, while only γ-austenite can be observed at 800 °C. From the above XRD spectra for the FeCoCr powders, it can be concluded that the variation of *M*_s_ during heat treatment is accompanied by the transformation among the γ-austenite, ε-martensite and α′-martensite phases. The transformation sequence may be γ→ε→α′, γ→α′ or ε→α′. Assuming that the value of *M*_s_ is proportional to the content of martensite phase and thorough transformation is achieved in the FeCoCr powders milled for 4 h and heat-treated at 500 °C, the conversion percentage of the martensite phase during ball milling would be roughly 61%, much lower than the 90% for a typical martensite conversion percentage in 304/316L stainless steel [31,33]. The reason for the low conversion percentage of the martensite phase in the FeCoCr powder during deformation may be due to the residual stress in particles which stabilized the austenite phase [34]. Nevertheless, after heat treatment, high content of the martensite phase in the FeCoCr powders was achieved due to further transformation of austenite and ε-martensite to α′-martensite [38], giving a higher *M*_s_ than that of 304/316L stainless steels. The *M*_s_–*T* curve of the FeCoCr powders milled for 4 h and heat-treated at 500 °C was also tested and shown in Appendix A. It can be seen that the FeCoCr powders exhibited high *T*_c_ and fairly stable *M*_s_ at temperature up to 500 °C, indicating their potential as efficient absorbents at high temperature.

The complex permittivity and permeability of the FeCoCr powders at different statuses were studied to verify the potential of deformation-thermal co-induced ferromagnetism for microwave absorption. Figure 3 shows the frequency-dependent characteristics of the complex permittivity ($\varepsilon_{r}=\varepsilon^{\prime }-i\varepsilon^{\prime \prime }$) and complex permeability ($\mu_{r}=\mu^{\prime }-i\mu^{\prime \prime }$) of the FeCoCr-paraffin composite over 1~18 GHz. The complex permittivity and permeability for the FeCoCr powders milled for 8 h and heat-treated at 500 °C were presented as typical examples. Overall, ball milling and subsequent heat treatment resulted in a large increase in ε′, ε″, $\mu^{\prime }$ and low-frequency *μ*″. Over the X band, with the increase in milling time from 0 to 8 h, ε′ increased from 6 to 80, while ε″ rose from less than 1 to higher than 10. The increase in electrical conductivity due to grain size growth is thought to be responsible for the increase in ε′ after heat treatment at 500 °C [39]. The increase in ε′ during ball milling and subsequent heat treatment was due to the enhancement of space charge polarization in the alternating electric field of the microwave since the specific surface area increased after milling. The increase in ε″ during ball milling and subsequent heat treatment was mainly due to the increase in inter-particle eddy current upon incidence of microwave [40] and the increase in intrinsic conductivity of the powders. This can be explained by the free-electron theory [41,42], i.e., $\varepsilon^{\prime \prime }=\sigma /2\pi f\varepsilon_{0}$, where σ is the electrical conductivity, *f* is the frequency, and $\varepsilon_{0}$ is the constant dielectric in vacuum. Therefore, ε″ increases with the extension of milling time due to an increase in conductivity in the FeCoCr-paraffin composite. When heat treatment was carried out for the milled FeCoCr powders, reduction of crystal defects such as intra-grain dislocation and grain boundaries occurred; this would weaken the scattering of electrons and enhance the intrinsic conductivity of the FeCoCr powders.

The complex permeability of the FeCoCr powders also exhibited a sharp increase after ball milling and heat treatment. Generally, *μ*′ showed a descending behavior with the increase of frequency because of ferromagnetic resonance and eddy current effect [43], while *μ*″ showed a typical resonant peak in the microwave range. With the increase in ball milling time and heat treatment temperature, *μ*′ in the low-frequency range increased rapidly. For instance, *μ*′ was only 1.13 for the raw FeCoCr powders at 1 GHz, while it increased to 2.45 after ball milling for 8 h and to 3.68 after subsequent heat treatment at 500 °C. A similar tendency was also found in the value of *μ*″ at resonant frequency (i.e., $\mu_{max}^{\prime \prime }$), which increased from 0.15 to ~1.40 after milling for 8 h and further increased to ~1.75 after heat treatment at 500 °C in a vacuum. Compared to the literature, our strategy can enable a large increase in *μ*″ since particle shape anisotropy was associated with phase transformation. Changing the shape of microwave absorbents from spherical to flaky was an efficient route to increase the permeability [44,45,46,47]. However, according to our best knowledge, a combination of phase transformation and purposely controlling shape anisotropy for microwave absorption has not been considered in previous studies on deformation-induced ferromagnetism. When deformation-induced ferromagnetism occurs in spherical particles, the initial permeability would be limited, since it generally follows an inverse relationship with its magnetic resonant frequency, i.e., the Snoek limitation [48]. For flaky absorbents, the behavior of permeability can go beyond the Snoek relationship, and it would exhibit a continuous increase upon deformation. With the employment of subsequent heat treatment, a further increase in permeability was achieved due to the increase in *M*_s_ and reduction of defects. On the other hand, the enhancement of complex permeability during subsequent heat treatment was due to the ferromagnetic phase transformation for an increase in *M*_s_ and the reduction of defects in powders. The magnetic loss capability of the FeCoCr powders is also enhanced due to the increase in *μ*″, see Appendix A, indicating its potential to attenuate the alternating magnetic field of the microwave. Therefore, high permeability was successfully obtained in our work by a combination of particle shape anisotropy, deformation-induced ferromagnetism and thermal-induced ferromagnetism.

To show the potential application of the FeCoCr powders for efficient microwave absorption at elevated temperature, electromagnetic parameters, microwave absorption and thermogravimetry spectra of the FeCoCr powders were further studied. When considering the microwave absorption, the impedance matching between the air and the composite consisting of the FeCoCr powders and the polymeric matrix is vitally important. Generally, impedance matching determines the amount of microwaves entering the materials [49]. In our study, even though high permeability can be achieved by deformation-thermal co-induced ferromagnetism, i.e., $\mu_{max}^{\prime \prime }$ of about 2.31 for the FeCoCr powders milled for 12 h and heat-treated at 500 °C in a vacuum, obtaining appropriate impedance matching is still the key issue. In Figure 4a, it can be seen that the complex permittivity of the FeCoCr powders was fairly high after ball milling for 12 h and heat treatment at 500 °C in a vacuum. The ε′ and ε″ in the X band reached 144~182 and 76~90, respectively; this lead to poor impedance matching and weak microwave absorption of the composite. To improve the impedance matching, the FeCoCr powders milled for 12 h were further heat-treated at 500 °C in the air to generate oxidation and insulate the surface to reduce the inter-particle eddy current upon incidence of the microwave. Compared to the FeCoCr powders heat-treated in a vacuum, those treated in air showed a significant reduction in complex permittivity, i.e., ε′ decreased to ~47, while ε″ reduced to ~9 over the X band. Even though the complex permeability of the FeCoCr powders decreased after heat treatment at 500 °C in the air, as shown in Figure 4b, the impedance matching of the material was still improved.

After the proper modulation of permittivity, the reflection loss (*RL*) of the FeCoCr powders was calculated and discussed. According to the transmission line theory, *RL* can be calculated by [50,51]
$$RL=-20\log\left|\frac{Z_{in}-Z_{0}}{Z_{in}+Z_{0}}\right|$$
$$Z_{in}=Z_{0}\sqrt{\frac{\mu_{r}}{\varepsilon_{r}}}\tanh\left(j\left(\frac{2\pi ft}{c}\right)\sqrt{\mu_{r}\varepsilon_{r}}\right)$$
where *Z*_0_ is the air impedance equal to 377 Ω, *Z_in_* is the input impedance of the absorbing material, *t* is the thickness of the absorbing material and c is the speed of light. From Figure 4c, it can be seen that the composite with the raw FeCoCr alloy powders has negligible microwave absorption capability, with a minimum *RL* of only −0.65 dB at thickness of 1 mm. The absorption capability of the FeCoCr powders was significantly enhanced after ball milling and heat treatment. However, when the FeCoCr powders were heat-treated in vacuum, the minimum *RL* for a 1 mm-thick composite was only −5 dB, due to its high permittivity and poor impedance matching. After heat treatment in air, the composite with the FeCoCr powders exhibited an *RL* of below −6 dB over 8–18 GHz and a minimum *RL* of −8.7 dB. The obvious enhancement of absorption indicates the potential of the deformation-thermal co-induced ferromagnetism strategy in producing materials with strong absorption performance. The absorption capability of the FeCoCr powder is lower than that of traditional carbonyl iron [52,53], due to the existence of the Cr element, which reduced the content of the magnetic elements. This also proves the applicability of utilizing austenite Fe-based alloy powders as microwave absorbents. Figure 4d exhibits the thermogravimetry spectra of traditional Fe and FeCoCr powders. It is obvious that oxidation of traditional carbonyl iron powders started at about 160 °C, while that of the FeCoCr powders started at about 300 °C, indicating their superior anti-oxidation property. The enhancement of anti-oxidation properties in FeCoCr was due to the existence of Co and Cr; the former is more difficult to oxidize than the Fe element [54], while the latter can form a dense protection layer for Fe elements [55]. From thermogravimetry results, the FeCoCr powders are expected to stay stable at 300 °C for long-term work and can be used for a short time at 500 °C.

## 4. Conclusions

In this work, strong microwave absorption was achieved for nanocrystalline austenite FeCoCr powders by deformation-thermal co-induced ferromagnetism. Via associating shape anisotropy during deformation, significant austenite-to-martensite transformation, as well as a considerable increase in both *M*_s_ and permeability, was obtained. The *M*_s_ and *μ*″ showed a parabolic behavior during deformation and further rapidly increased with subsequent heat treatment. After milling for 4 h, the *M*_s_ increased from 1.43 to about 109.92 emu/g, along with the formation of a flaky morphology and rise of $\mu_{max}^{\prime \prime }$ from about 0.15 to 0.92. The *M*_s_ and $\mu_{max}^{\prime \prime }$ further increased to 181.58 emu/g and 2.31 after subsequent heat treatment at 500 °C. Strong microwave absorption of higher than 6 dB over 8~18 GHz at a thickness of 1 mm was obtained in the resultant FeCoCr powders, indicating their potential application as high efficient microwave absorbents at elevated temperatures and in corrosive environments.

Thus, by taking the FeCoCr austenite powder as an example, we developed a route to utilize austenite Fe-based alloys for microwave absorption. This can largely broaden the potential members of absorbents and integrate more functions into absorbents such as superior mechanical traits, anti-oxidation, anti-corrosion, etc. Moreover, this can relieve the reliance on the crystal structure of resultants during gas atomization on the production technique parameters, such as cooling rate and production environment, since the austenite phase can totally transform into the martensite phase by deformation-thermal co-induced ferromagnetism. Additionally, our strategy is easy to use with simple apparatus, which can save on cost, energy and reduce carbon emissions during industrial production.

## Figures and Tables

**Figure 1 nanomaterials-12-02263-f001:**
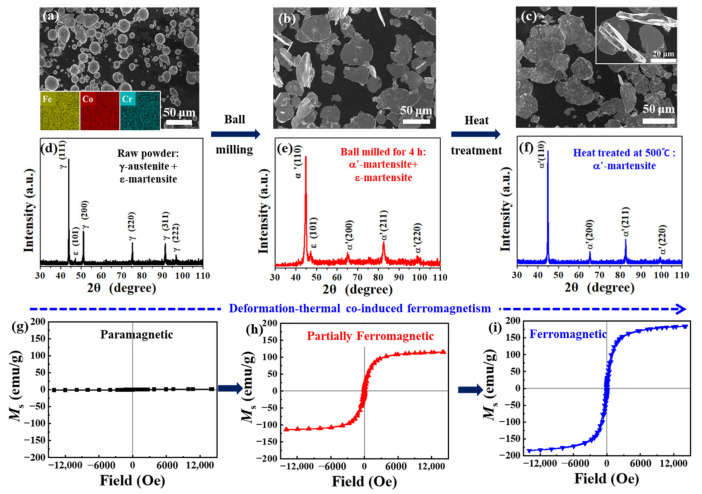
SEM images, XRD spectra and hysteresis loops of (**a**,**d**,**g**) raw FeCoCr powders, (**b**,**e**,**h**) those after ball milling for 4 h, and (**c**,**f**,**i**) subsequent heat treatment at 500 °C. Insets in (**a**) is the energy dispersive X-ray spectroscopy images for element distribution, while that in (**c**) is the SEM image for lateral view of FeCoCr powders.

**Figure 2 nanomaterials-12-02263-f002:**
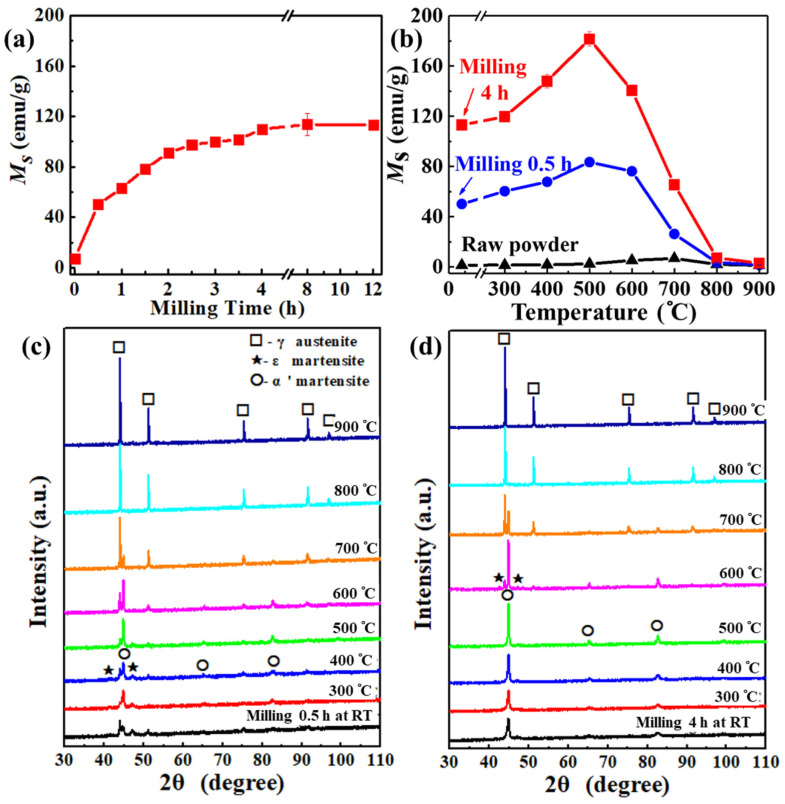
Evolution of *M*_s_ in FeCoCr powders during (**a**) ball milling and (**b**) subsequent heat treatment; XRD spectra of FeCoCr powders (**c**) after ball milling for 0.5 h and subsequent heat treatment and (**d**) after ball milling for 4 h and subsequent heat treatment.

**Figure 3 nanomaterials-12-02263-f003:**
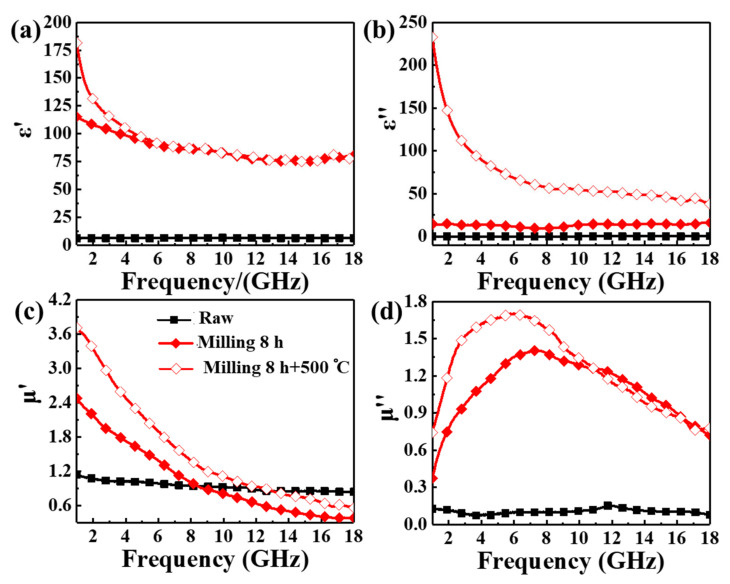
Variation of (**a**,**b**) complex permittivity and (**c**,**d**) complex permeability for FeCoCr powders during deformation-thermal co-induced ferromagnetism.

**Figure 4 nanomaterials-12-02263-f004:**
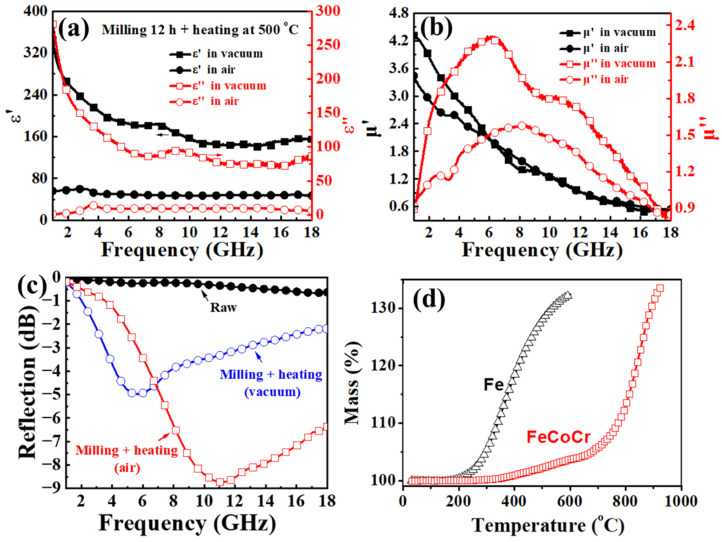
(**a**) Complex permittivity, (**b**) complex permeability and (**c**) simulated *RL* of FeCoCr powders after ball milling and subsequent heat treatment in vacuum or air; (**d**) Thermogravimetry spectra of Fe and FeCoCr powders.

## Data Availability

The data presented in this study are available on request from the corresponding author.

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
