# Peer review of "Deformation-Thermal Co-Induced Ferromagnetism of Austenite Nanocrystalline FeCoCr Powders for Strong Microwave Absorption"

_nanomaterials, 2022, doi:10.3390/nano12132263_

Round 1
Reviewer 1 Report
The manuscript, represent an interesting approach in the application of the designed material – for microwave absorption. The material is not new, but the approach represents an aspect less considered up to now. I, believe that the work will have a good impact in the field.
The English level is good.
Reviewer 2 Report
The manuscript by Fu et. al, demonstrates ferromagnetism and strong microwave absorption in FeCoCr powders. The manuscript is well written and should be accepted after minor revision.
When the average particles size or flakes size is in the range of 50μm -200 μm, we should not say these are nanocrystals or nanoparticles. I would suggest the authors explain why they considered the materials as nanocrystalline powders. If the thickness of fakes (Fig 1c) is sub-100 nm, then they should include the corresponding supporting data. I also did not get why the size was increasing after ball milling.
Magnetization vs. temperature data (MT curves) may be included to determine the Tc for 0.5 and 4h milling samples. This may provide more information than what they got from Figure 2b.
Author Response
We appreciate the positive comment of the reviewer for our manuscript, which we believe can encourage us for future work.
Reviewer 3 Report
Within the article, the authors presented how thermal deformation of FeCoCr nanocrystals (austenite phase) causes the material's ferromagnetic properties and enhances thereby strong microwave adsorption. I have no specific concerns about the article. Data is correctly presented and support the author's claims.
The research topic is probably of average significance and soundness however, it might be useful for scientists working in this area. Thus, I think that the article can be published in the Nanomaterials journal
